DATA RELEASE

# Genome assembly and annotation of the Brown-Spotted Pit viper *Protobothrops mucrosquamatus*

Xiaotong Niu[1,2,†], Haorong Lu[3,4,†], Minhui Shi[1,5], Shiqing Wang[1,5], Yajie Zhou[1,4,*] and Huan Liu[1,4,*]

1 State Key Laboratory of Agricultural Genomics, BGI-Shenzhen, Shenzhen, 518083, China
2 School of Ecology, Sun Yat-sen University, Shenzhen, 510275, China
3 BGI-Shenzhen, Shenzhen, 518083, China
4 China National GeneBank, BGI-Shenzhen, Shenzhen, 518120, China
5 College of Life Sciences, University of Chinese Academy of Sciences, Beijing, 100049, China

## ABSTRACT

The Brown-Spotted Pit viper (*Protobothrops mucrosquamatus*), also known as the Chinese habu, is a widespread and highly venomous snake distributed from Northeastern India to Eastern China. Genomics research can contribute to our understanding of venom components and natural selection in vipers. Here, we collected, sequenced and assembled the genome of a male *P. mucrosquamatus* individual from China. We generated a highly continuous reference genome, with a length of 1.53 Gb and 41.18% of repeat elements content. Using this genome, we identified 24,799 genes, 97.97% of which could be annotated. We verified the validity of our genome assembly and annotation process by generating a phylogenetic tree based on the nuclear genome single-copy genes of six other reptile species. The results of our research will contribute to future studies on *Protobothrops* biology and the genetic basis of snake venom.

**Subjects** Genetics and Genomics, Evolutionary Biology, Zoology

Submitted: 22 May 2023

\* Corresponding authors. E-mail:
zhouyajie@genomics.cn;
liuhuan@genomics.cn

† Contributed equally.

Preprint submitted at https://doi.org/10.20944/preprints202310.1073.v1

Included in the series: *Snake Genomes* (https://doi.org/10.46471/GIGABYTE_SERIES_0004)

## INTRODUCTION

*Protobothrops mucrosquamatus* belongs to the Viperidae (viper) family of snakes commonly known as the brown spotted pit viper or Chinese habu. This species is widely distributed in northern Vietnam, Laos, northern Myanmar, northeastern India, as well as southwestern and eastern China (Figure 1) [1]. *P. mucrosquamatus* is a venomous snake with tubular venom-conducting fangs and loreal pit. Their poisoning manifests through the functional impairment of the blood circulation system of their prey [2]. Compared with other terrestrial vipers, the maximum amount of single-discharging venom of *P. mucrosquamatus* is higher than in *Trimeresurus stejnegeri*, *Gloydius blomhoffii* and *Bungarus multicinctus* [3]. Its toxicity per unit dose is also higher than in *Deinagkistrodon acutus* and *T. stejnegeri* [3].

Snake venom, while it may contribute to health damage in organisms [1, 2, 4–6], can also play a role in biomedicine [5, 7–9], particularly in snake antivenom development, disease treatment and many other fields [10]. High-quality reference genomes and transcriptomes are required to detect venom genes, get insights into toxin-manufacturing mechanisms, and design safe and effective antivenoms and other drugs [11, 12]. Moreover, the rapid evolution of venom proteins generally occurs under environmental stress [13, 14], such as predation needs. Hence, the study of proteinaceous-venom coding genes is an excellent model system for adaptation and nature selection [15].

**Figure 1.** A Brown-Spotted Pit viper (*P. mucrosquamatus*) individual, photographed by Diancheng Yang in Guilin, Guangxi Province.

## MAIN CONTENT

### Context

While snake venoms are dangerous to human health, they are also a potential gold mine of bioactive proteins that can be harnessed for drug discovery [16]. Also, snake genomics has huge potential for studying venom evolution and toxicology. Here, we assembled a highly contiguous genome of a male *P. mucrosquamatus* individual collected from Guilin, Guangxi, China, using single-tube long fragment read (stLFR) technology [17] and whole genome sequencing (WGS). The total size of the genome we generated is 1.53 Gb, including 41.18% repeat elements. This data provides new material for future research on the *Protobothrops* genome and the genetic basis of this snake venom.

### Methods

Detailed stepwise protocols are gathered in a protocols.io collection, with the minor adaptations outlined below [18] (Figure 2).

### Sample collection and sequencing

A male *P. mucrosquamatus* individual was captured in Guilin, Guangxi, China. After collection and identification, the specimen was quickly frozen in −80 °C Drikold dry ice for storage and transport in order to preserve DNA and RNA molecules. Samples from the heart, stomach, liver, and kidney were utilized for RNA sequencing. A muscle sample was used for stLFR and WGS sequencing. DNA extraction, library construction and sequencing are outlined in the protocols.io protocols [18].

The Institutional Review Board of BGI (BGI-IRB E22017) approved sample collection, experiments, and research design in this study. Throughout this research, strict adherence to the guidelines set by BGI-IRB was ensured during all procedures.

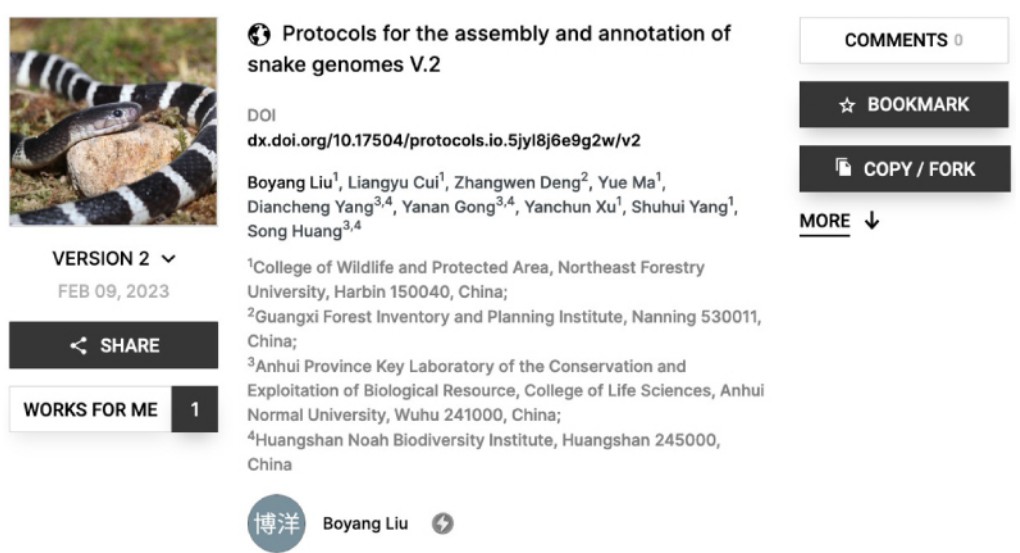

**Figure 2.** Protocols.io collection of the standard protocols for sequencing snake genomes [18].
https://dx.doi.org/10.17504/protocols.io.4r3l27ez4g1y/v1

## Genome assembly, annotation and assessment

Supernova software (v2.1.1; RRID:SCR_016756) was employed to assemble the stLFR sequencing data. To address any gaps and eliminate redundancies in this assembly, the WGS data was subjected to gap filling and redundancy removal using GapCloser [19] (v1.12-r6; RRID:SCR_015026) and redundans (v0.14a) [20], respectively.

In order to identify known repeat elements in genome sequences, a combination of tools was utilized: Tandem Repeat Finder [21] (v. 4.09), LTR_Finder (RRID:SCR_015247) [22], RepeatModeler [23] (v1.0.8; RRID:SCR_015027), RepeatMasker [24] (v. 3.3.0; RRID:SCR_012954) and RepeatProteinMask (v. 3.3.0) [25]. For the prediction of protein-coding genes, multiple approaches were employed. *De novo* gene prediction was performed using Augustus (v3.0.3; RRID:SCR_008417) [26]. The RNA-seq data was filtered with Trimmomatic (v0.30; RRID:SCR_011848) [27]. Then, the transcript assembly was performed using Trinity (v2.13.2; RRID:SCR_013048) [28] and based on clean RNA-seq data. Alignment of transcripts against the genome to obtain gene structures was performed using Program to Assemble Spliced Alignments (or PASA) (v2.0.2; RRID:SCR_014656) [29]. Homology-based prediction involved mapping protein sequences from the UniProt database (release-2020_05) of *Pseudonaja textilis*, *Thamnophis elegans* and *Notechis scutatus* to the genome using the Blastall (v2.2.26) [30] with an *E*-value cut-off of $1 \times 10^{-5}$. Gene models were predicted by analyzing the alignment results with GeneWise [31] (v2.4.1; RRID:SCR_015054). Integration of RNA-seq, homology, and *de novo* predicted genes to generate the final gene set was achieved using the MAKER pipeline (v3.01.03; RRID:SCR_005309) [32].

To annotate the function of genes of *P. mucrosquamatus*, a comprehensive analysis was conducted. BLAST searches were executed against multiple databases, including SwissProt, TrEMBL (RRID:SCR_004426), and Kyoto Encyclopedia of Genes and Genomes (KEGG; RRID:SCR_012773), with an *E*-value cut-off of $1 \times 10^{-5}$. To predict motifs and domains, InterProScan (v5.52-86.0; RRID:SCR_005829) [27] as well as gene ontology



**Table 1.** Summary statistics of *P. mucrosquamatus* sequenced reads.

|  |  | Base number | GC content (%) | Q20 (%) | Q30 (%) |
|---|---|---|---|---|---|
| WGS | fq1 | 52,036,970,400 | 40.30 | 97.58 | 92.48 |
|  | fq2 | 52,036,970,400 | 40.23 | 97.98 | 92.71 |
| stLFR | fq1 | 104,698,910,600 | 38.89 | 96.9 | 90.75 |
|  | fq2 | 136,108,583,780 | 41.72 | 97.79 | 91.85 |

**Table 2.** Summary of the features of the *P. mucrosquamatus* genome.

| Statistical level | Original | | | Scaffold > (500) bp | |
|---|---|---|---|---|---|
|  | Scaffold | Contig | Contig > (500) | Scaffold | Contig |
| Total number (>) | 203,555 | 287,462 | 192,124 | 149,173 | 232,200 |
| Total length of (bp) | 153,064,8812 | 1,481,196,605 | 1,457,896,424 | 1,512,499,815 | 1,463,075,630 |
| Average length (bp) | 7,519.58 | 5,152.67 | 7,588.31 | 10,139.23 | 6,300.93 |
| N50 Length (bp) | 380,005 | 36,547 | 37,585 | 390,274 | 37,334 |
| N90 Length (bp) | 2,960 | 2,304 | 2,773 | 3,453 | 2,667 |
| Maximum length (bp) | 5,566,463 | 488,153 | 488,153 | 5,566,463 | 488,153 |
| GC content (%) | 39.86 | 39.86 | 39.79 | 39.8 | 39.8 |

(GO; RRID:SCR_002811) were employed. The results of this analysis further enriched our understanding of the genes' roles and their involvement in biological processes.

The completeness of the genome was evaluated using sets of Benchmarking Universal Single-Copy Orthologs (BUSCO; v5.2.2; RRID:SCR_015008) with genome mode and lineage data from vertebrata_odb10 [33]. To reconstruct the phylogenetic tree, we used OrthoFinder (v2.3.7; RRID:SCR_017118) [34] to search for single-copy orthologs among the protein sequences of *Anolis carolinensis* (GCA_000090745.2), *Chelonia mydas* (GCA_015237465.2), *Danio rerio* (GCA_000002035.4), *Deinagkistrodon acutus* [35], *Gallus gallus* (GCA_016699485.1), *Homo sapiens* (GCA_000001405.29), *Mus musculus* (GCA_000001635.9), *Ophiophagus hannah* (GCA_000516915.1), *Python bivittatus* (GCA_000186305.2), *Xenopus tropicalis* (GCA_000004195.4) and *Alligator mississippiensis* (GCA_000281125.4).

## RESULTS

In this snake genomics study, 224.27 Gb linked-reads data was obtained after stLFR sequencing, and 96.93 Gb short reads data was obtained after WGS sequencing, for a total of 321.20 Gb (Table 1).

We produced a high-continuity *P. mucrosquamatus* genome assembly, with 1.53 Gb total genome size, 39.86% GC content and 362.40 kb scaffold N50 length (Table 2). The *P. mucrosquamatus* genome assembly, whose maximal scaffold length reaches 5.31 M, has 149,173 scaffolds over 500 bp, with 1.51 Gb total length, occupying 98.82% of the entire genome. We foresee that this resource will provide new perspectives for the study of viper genomics.

We identified 41.18% repetitive elements in our *P. mucrosquamatus* genome. Long interspersed nuclear elements (LINEs) constituted the largest proportion of this assembly at 32.33%, equivalent to 471.99 Mb. This figure is very similar to the repetitive element content in a previously sequenced *Thamnophis elegans* genome (42.02%) (accession No. PRJNA561996) and *Crotalus tigris* genomes (42.31%) [36], indicating consistency in the observed values. The other dominant examples of transposable elements (TEs), LTRs, DNA transposons and SINE were 11.50%, 4.94% and 0.80%, respectively (Figure 3, Tables 3 and 4).

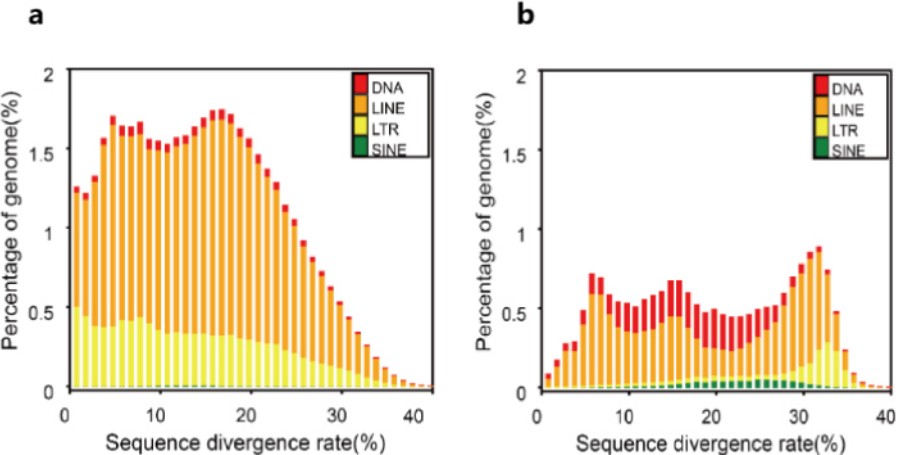

**Figure 3.** Distribution of TEs in our *P. mucrosquamatus* genome. The TEs include DNA transposons (DNA) and RNA transposons (i.e., DNAs, LINEs, LTRs and SINEs). (a) Distribution of *de novo* sequence divergence rates. (b) Distribution of known sequence divergence rates.

**Table 3.** Statistics for the repetitive sequences identified in our *P. mucrosquamatus* genome.

| Type | Repeat size | % of genome |
|---|---|---|
| Trf | 48,630,912 | 3.177144 |
| Repeatmasker | 248,960,159 | 16.265008 |
| Proteinmask | 178,699,911 | 11.674782 |
| *De novo* | 591,205,406 | 38.624497 |
| Total | 630,311,866 | 41.179391 |

**Table 4.** Summary of the TEs in our *P. mucrosquamatus* genome.

| Type | Repbase TEs | | TE proteins | | *De novo* | | Combined TEs | |
|---|---|---|---|---|---|---|---|---|
| | Length (bp) | % in genome | Length (bp) | % in genome | Length (bp) | % in genome | Length (bp) | % in genome |
| DNA | 54,802,686 | 3.580357 | 2,721,607 | 0.177807 | 23,812,202 | 1.555693 | 75,566,775 | 4.936911 |
| LINE | 173,499,745 | 11.335046 | 145,892,994 | 9.531448 | 446,008,208 | 29.138507 | 494,919,112 | 32.333943 |
| SINE | 11,128,833 | 0.727066 | 0 | 0 | 1,414,004 | 0.092379 | 12,299,674 | 0.80356 |
| LTR | 27,382,417 | 1.788942 | 30,199,813 | 1.973007 | 165,177,572 | 10.791344 | 175,979,322 | 11.497041 |
| Other | 95,860 | 0.006263 | 0 | 0 | 0 | 0 | 95,860 | 0.006263 |
| Total | 248,960,159 | 16.265008 | 178,699,911 | 11.674782 | 588,493,585 | 38.447329 | 618,611,286 | 40.414972 |

Using homology-based, *de-novo* and RNA-sequencing annotation methods, 24,799 protein-coding genes were identified in our *P. mucrosquamatus* genome assembly. The average gene of a *P. mucrosquamatus* is 1.53 bp long and contains 8.96 exons. Additionally, 387 miRNAs, 319 tRNAs and 289 snRNAs were predicted in our *P. mucrosquamatus* genome (Table 5).

Through comparisons with public datasets, including InterPro [37], KEGG [38], SwissProt [39], TrEMBL [39] and GO terms, 24,296 expanded gene families were identified, and 97.97% of genes could be annotated based on their function (Table 6).

According to our KEGG enrichment analysis, Environmental Information Processing, Organismal Systems and Metabolism pathways comprise a significant proportion of these pathways. In particular, the Signal Transduction pathways take up the largest proportion. Genes associated with the Immune (2,445) and Endocrine systems (2,033) accounted for the

**Table 5.** Statistics for the miRNA, tRNA, rRNA and snRNA predicted in our *P. mucrosquamatus* genome.

| Type | | Copy (w) | Average length (bp) | Total length (bp) | % of genome |
|---|---|---|---|---|---|
| miRNA | | 387 | 115.3540052 | 44,642 | 0.002917 |
| tRNA | | 319 | 76.38244514 | 24,366 | 0.001592 |
| rRNA | rRNA | 75 | 111.8266667 | 8,387 | 0.000548 |
| | 18S | 18 | 141.5555556 | 2,548 | 0.000166 |
| | 28S | 52 | 104.3269231 | 5,425 | 0.000354 |
| snRNA | snRNA | 289 | 115.6955017 | 33,436 | 0.002184 |
| | CD-box | 110 | 90.2 | 9,922 | 0.000648 |
| | HACA-box | 66 | 144.7575758 | 9,554 | 0.000624 |
| | splicing | 98 | 112.1734694 | 10,993 | 0.000718 |

**Table 6.** Results of gene functional annotation.

| Values | Total | Swissprot-Annotated | KEGG-Annotated | TrEMBL-Annotated | Interpro-Annotated | GO-Annotated | Overall |
|---|---|---|---|---|---|---|---|
| Number | 24,799 | 21,141 | 21,203 | 23,741 | 23,579 | 15,322 | 24,296 |
| Percentage | 100% | 85.25% | 85.50% | 95.73% | 95.08% | 61.78% | 97.97% |

largest number of Organismal System pathways (Figure 4a). Based on our GO analysis, 7,900 genes relate to binding and 7,740 genes to cellular processes (Figure 4b).

## DATA VALIDATION AND QUALITY CONTROL

BUSCO v5.2.2 was used to evaluate the completeness and quality of our assembly [40]. Our BUSCO analysis results indicate that this genome assembly has up to 83.6% completeness using the vertebrata_odb10 database (Figure 5).

To check the quality of our assembly, we constructed a phylogenetic tree using protein sequences from NCBI and CNGB for seven other kinds of amphibians and reptiles (*Anolis carolinensis*, *Chelonia mydas*, *Deinagkistrodon acutus*, *Ophiophagus hannah*, *Python bivittatus*, *Xenopus tropicalis* and *Alligator mississippiensis*), as well as *Gallus gallus*, *Homo sapiens*, *Mus musculus*, *Danio rerio*. The relationship among all these species reflected by the phylogenetic tree aligns with previous research, demonstrating that our data can screen related species (Figure 6). Finally, a total of 1,177 single-copy loci were found.

## REUSE POTENTIAL

This genomic data will provide new resources for further studying viper biology and evolution alongside the genetic basis of viper snake venom.

## DATA AVAILABILITY

The data that support the findings of this study have been deposited into the CNGB Sequence Archive (or CNSA) [41] of China National GeneBank DataBase (or CNGBdb) [42] with the accession number CNP0004048. Raw reads are available in the Short Read Archive under the BioProject ID PRJNA943598, and additional data is available in the GigaDB repository [43].

## EDITOR'S NOTE

This paper is part of a series of Data Release papers presenting the reference genomes of different snake species [44].

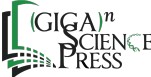

**Figure 4.** Gene annotation information of *P. mucrosqamatus*. (a) KEGG enrichment of *P. mucrosquamatus*. (b) GO enrichment of *P. mucrosquamatus*. (c) Venn diagram of InterPro, KEGG and Swissport annotation results.

## ABBREVIATIONS

BGI-IRB, Institutional Review Board of BGI; BUSCO, Benchmarking Universal Single-Copy Orthologs; GO, gene ontology; KEGG, Kyoto Encyclopedia of Genes and Genomes; LINE, long interspersed nuclear element; LTR, long terminal repeat; SINE, short interspersed nuclear

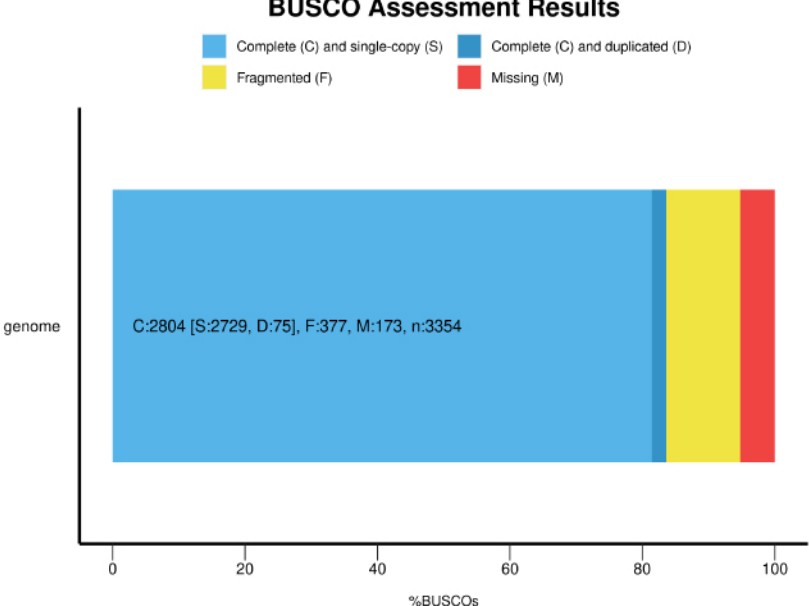

**Figure 5.** BUSCO assessment result of our *P. mucrosquamatus* genome.

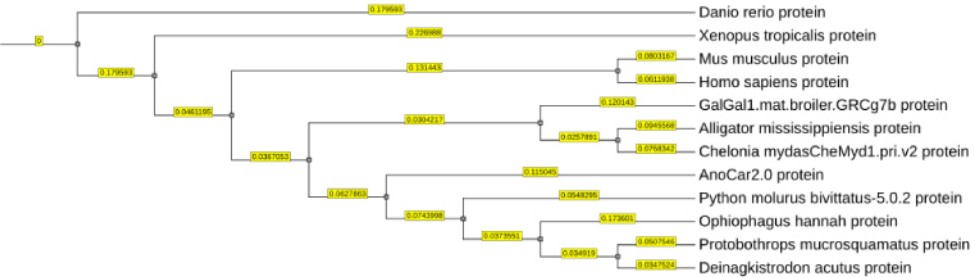

**Figure 6.** Phylogenetic tree reconstructed using single-copy genes from nuclear genomes. The numbers on the branches of the phylogenetic tree represent the branch length obtained in OrthoFinder.

element; stLFR, single-tube long fragment read; TE, transposable element; WGS, whole genome sequencing.

## DECLARATIONS

### Ethics approval and consent to participate
The authors declare that ethical approval was not required for this type of research.

### Competing interests
The authors declare no conflict of financial interests.

## Authors' contributions

H Liu designed and initiated the project. H Lu, YZ and MS performed the DNA extraction and the library construction. XN and SW performed the data analysis and wrote the manuscript. All authors read and approved the final manuscript.

## Funding

Our project was supported by the China National GeneBank (or CNGB) and the Guangdong Provincial Key Laboratory of Genome Read and Write (grant no. 2017B030301011). This work was also supported by BGI-Shenzhen.

## Acknowledgement

Anhui Normal University collected the samples.

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
