## [Editor Report]

Editor’s AssessmentThe Brown-Spotted Pit viper Protobothrops mucrosquamatus, also known as the Chinese habu, is a widespread and highly venomous snake distributed from from NE India to Eastern China. To help better understand the evolution of pit vipers, a 1.53 Gb reference genome was sequenced, assembled and described in this work. During review some inconsistencies the metrics were fixed. This data can be combined with already published and upcoming snake genome data to construct the evolutionary history of snakes and other reptiles as well as the genetic basis of snake venom.

---

## [Reviewer Report]

Reviewer name and names of any other individual's who aided in reviewer Yasuhiro GoDo you understand and agree to our policy of having open and named reviews, and having your review included with the published papers. (If no, please inform the editor that you cannot review this manuscript.)YesIs the language of sufficient quality?YesPlease add additional comments on language quality to clarify if needed
Are all data available and do they match the descriptions in the paper? YesAdditional CommentsAre the data and metadata consistent with relevant minimum information or reporting standards? See GigaDB checklists for examples <a href="http://gigadb.org/site/guide" target="_blank">http://gigadb.org/site/guide</a>YesAdditional CommentsIs the data acquisition clear, complete and methodologically sound?YesAdditional CommentsIs there sufficient detail in the methods and data-processing steps to allow reproduction?YesAdditional CommentsIs there sufficient data validation and statistical analyses of data quality? YesAdditional CommentsIs the validation suitable for this type of data?YesAdditional CommentsIs there sufficient information for others to reuse this dataset or integrate it with other data?YesAdditional CommentsAny Additional Overall Comments to the Author1. The value of repeat element content is 41.18% in the Abstract, but the Main Content value is 38.62%, which is inconsistent with the Abstract value. I would like to see the values be unified into one (Total value?).  2. Figure 1 should show not only a picture of the snake but also its distribution area (habitat).  3. The first sentence of the Result states "224.27 Gb long reads data," but single-tube long fragment read (stLFR) is not a true long read. The term "linked-read" is better.  4. Tables 3 and 4 do not have specific descriptions of "De novo," so please provide more details.  5. The authors use BUSCO to evaluate gene completeness, but I recommend trying compleasm (https://github.com/huangnengCSU/compleasm), a recently improved version of BUSCO.  6. The animals in the parentheses after "For the purpose of checking the quality of our assembly, six other kinds of amphibians and reptiles" in the "Data validation and quality control" section also use animals other than amphibians and reptiles, so please correct the sentence appropriately.  7. Figure 5C needs to be explained in the text.  8. There is no explanation of the meaning of the numbers in the branches of the phylogenetic tree in Figure 7. There needs to be an explanation of how they were obtained.
RecommendationMinor Revision

---

## [Reviewer Report]

Reviewer name and names of any other individual's who aided in reviewer Yan ChaochaoDo you understand and agree to our policy of having open and named reviews, and having your review included with the published papers. (If no, please inform the editor that you cannot review this manuscript.)YesIs the language of sufficient quality?NoPlease add additional comments on language quality to clarify if needed
A complete English revision is requiredAre all data available and do they match the descriptions in the paper? YesAdditional CommentsI did not find description of method for RNA-Seq in manuscript.Are the data and metadata consistent with relevant minimum information or reporting standards? See GigaDB checklists for examples <a href="http://gigadb.org/site/guide" target="_blank">http://gigadb.org/site/guide</a>YesAdditional CommentsIs the data acquisition clear, complete and methodologically sound?YesAdditional CommentsIs there sufficient detail in the methods and data-processing steps to allow reproduction?YesAdditional CommentsIs there sufficient data validation and statistical analyses of data quality? YesAdditional CommentsIs the validation suitable for this type of data?YesAdditional CommentsIs there sufficient information for others to reuse this dataset or integrate it with other data?YesAdditional CommentsCareful language modifications are needed, particularly in the abstract and introduction sections. Here are some detailed suggestions.  1. "In recent years, P. mucrosquamatus has generated a number of poisonous snake bite cases in southeastern China.". This does not serve as an optimal starting point for the research. To enhance its impact, consider incorporating more specific statistical data, such as the annual number of people bitten or killed by the snake species.  2. "Genomics research can provide much insight in understanding toxin-production mechanisms and natural selection in vipers. ". The term 'toxin-production mechanisms' suggests an intricate process that the genome typically does not reveal directly; instead, it usually provides information about venom components rather than their underlying mechanisms. Additionally, the scope of the following study does not encompass toxin genes and natural selection. The author may consider prioritizing data publication or incorporating relevant analyses to strengthen the research. 3. " Here, we collected a male P.... 41.18% repeat element content." . This sentence is excessively lengthy, requiring revision for improved readability. 4. "97.97% genes could be annotated based on function". The statement 'annotated based on function' is not suitable in this context. 5. "venom‐conducting fangs and cheek fossa", 'cheek fossa' should be 'loreal pit'.  6. "Compared with other terrestrial vipers ...Gloydius blomhoffii and Bungarus multicinctus" need references. 7. " making the study of proteinaceous-venoms coding genes an excellent model system for the study of adaptation and nature selection". model system refer to what? study or venoms? 8. Figure 2: Including a screenshot here is unnecessary; providing the reference should suffice. 9. "After collection and identification, the specimen was quickly frozen in -80°C drikold dry ice during storage and transport in order to maintain high quality for further use. " I am somewhat confused about the tissue used here. Additionally, the author should provide information about the Animal Ethics Committee agreement or approval. 10. "the RNAseq data underwent filtration with Trimmomatic“, However, there is a lack of information on how the RNAseq was performed. Please include details about the RNAseq methodology. 11. "We produced a high-continuity P. mucrosquamatus genome assembly, with 1.53Gb total genome size,". A scaffold N50 below 500k indicates a genome with lower continuity. Further efforts are needed to improve the genome's quality. Additionally, consider providing information on the genome size calculated by the kmer method. 12. "For the purpose of checking the quality of our assembly, " I would recommend comparing quality indicators, such as N50 and Gene Number, with those of other genomes.Any Additional Overall Comments to the AuthorThis study presents the genome of the venomous snake Protobothrops mucrosquamatus, utilizing stLFR and WGS data. The research further assesses the genome's quality and performs repeat and gene annotation. As this journal prioritizes data publication, an evaluation of innovativeness is not conducted. However, the current genome quality does not meet modern standards, further efforts are needed to improve the genome's quality. Moreover, It would be beneficial to conduct an analysis of the toxin genes detected in the genome.RecommendationAccept